# Optimization of the Semi-Active-Suspension Control of BP Neural Network PID Based on the Sparrow Search Algorithm

**DOI:** 10.3390/s24061757

**Published:** 2024-03-08

**Authors:** Mei Li, Jie Xu, Zelong Wang, Shuaihang Liu

**Affiliations:** Mechanical and Electrical Engineering College, Hainan University, Haikou 570100, China; 21220951360012@hainanu.edu.cn (J.X.); 22220854060044@hainanu.edu.cn (Z.W.); 23220854060019@hainanu.edu.cn (S.L.)

**Keywords:** hub motor, neural network, sparrow search algorithm, PID control, semi-active suspension, vehicle smoothness

## Abstract

Electric vehicles with hub motors have integrated the motor into the wheel, which increase the unsprung mass of the vehicle, and intensifies the vibration of the underspring components. The motor excitation during driving also intensifies the wheel vibration. The coupling effect between the two makes the performance of electric vehicles deteriorate. The article employed a disc-type permanent-magnet motor as the hub motor, taking into consideration the increase in sprung mass caused by the hub motor and the adverse effects of vertical vibration from motor excitation. Based on random road-surface excitation, and considering the secondary excitation caused by wheel motor drive and vehicle-road coupling, a coupled-dynamics model of a semi-active-suspension vehicle-road system for vertical vehicle motion is investigated under multiple excitations. Using body acceleration, suspension deflection, and dynamic tire load as evaluation indicators, a BP neural network PID controller based on the sparrow search algorithm optimization is proposed for the semi-active-suspension system. Compared with PID control and particle swarm optimization (PSO-BPNN-PID), the research findings indicate that the optimized semi-active suspension significantly improves the ride comfort of hub-motor electric vehicles, and meets the requirements for control performance under different vehicle driving conditions.

## 1. Introduction

In recent years, environmental protection and energy conservation have been the focus of attention in the automotive industry in various countries around the world [1,2]. The electric vehicle (EV) driven by hub motors has gained widespread attention due to its high transmission efficiency and ease of implementing various intelligent controls. However, driving by installing hub motors inside the wheel hub of an EV is a more centralized driving method, but it increases the vehicle’s underspring component mass and exacerbates the vibration of the underspring components. Furthermore, the electromagnetic excitation generated by the hub motors during driving affects the vertical vibration of the vehicle, and the secondary excitation caused by the vehicle vibration acting on the road surface further affects the ride comfort of the vehicle. Therefore, it is necessary to study the vehicle–road coupling of the suspension system for hub-motor EVs with a focus on their vertical dynamic characteristics [3].

The dynamics of vehicle–road interaction have received great attention from researchers over the past decade [4,5,6,7]. In traditional road dynamics, vehicles are considered moving loads excited by the roughness of the road surface, with little consideration for the influence of road vibrations [8]. Therefore, it is necessary to conduct a coupled dynamic analysis of the vehicle–road system. Hu [4] studied the nonlinear coupled dynamics of the vehicle–road system by modeling the road as a Timoshenko beam on a nonlinear foundation to investigate the dynamic response of the coupled vehicle–road system. This study considered the shear deformation of the road and subgrade, the shear modulus of the beam, and the shear deformation coefficient of the subgrade, and conducted a numerical analysis of the coupling effect between the road and the vehicle using convergent modal truncation, comparing the physical parameters of the vehicle–road system with the coupled-vibration response. Zheng [5] investigated the dynamic response of highway embankments under quasi-static and dynamic loads using a vehicle-road-ground coupling model. A multi-degree-of-freedom vehicle system was established, and dynamic wheel–road forces were considered by introducing Hertz contact springs between the wheels and the road surface. The study examined the effects of vehicle speed, road roughness parameters, and wheel–road contact stiffness on dynamic wheel–road forces and embankment response. Snehasagar [6] studied the effect of a viscoelastic road-surface model on the dynamics of the vehicle–road coupling system. The Galerkin method and the Runge–Kutta method were used to discretize the differential equations generated by the system’s motion dynamics equation. The study examined the effects of road roughness, vehicle acceleration, and temperature on system response, as well as the effects of coupling on road displacement and vehicle vertical displacement. Li [7] proposed a nonlinear vehicle–road coupling model that considered the nonlinearity of suspension stiffness, suspension damping, and tire stiffness, as well as the viscoelasticity of the asphalt surface layer. The simulation results of the coupling model were compared with those of the traditional non-coupling model. The proposed nonlinear vehicle–road coupling model will lead to higher computational accuracy and enable the simultaneous design of both vehicles and roads. Krishnanunni [9] proposed an iterative decoupling technique for analyzing the dynamic response of the vehicle–road system. Two road models, a Timoshenko beam on a nonlinear foundation and a double-layer rectangular thin plate, were analyzed for their nonlinear foundation and the effect of coupling on road displacement and vehicle vertical displacement, emphasizing the dynamic interaction between the vehicle and road. To address the dynamic-coupling effects during vehicle operation, it is also essential to integrate the road-surface excitations during the vehicle’s motion (random road-surface excitations) and dynamic-coupling road-surface excitations (road-surface secondary excitation).

Various types of motors, such as induction motors, switched reluctance motors, and permanent magnet synchronous motors [10], can be used in electric vehicles. During vehicle operation, the motors internally induce vibrations and generate some unbalanced electromagnetic forces, which are directly transmitted to the wheels, as a result, exacerbate wheel vibration and affect vehicle ride comfort. Researchers have proposed various suspension systems for motor excitation [11]. Shao [12] analyzed the coupling effect of road excitation and wheel-mounted switched reluctance motors (SRMs) on vehicle ride comfort, and proposed a hybrid control system consisting of a fault-tolerant H∞ suspension controller and an SRM controller. By using a combination of current chopping control (CCC) and pulse width modulation (PWM) to adjust the SRM controller, the proposed hybrid control method can effectively reduce SRM eccentricity and residual unbalanced radial force, achieving better vehicle ride comfort. Wu [13] established an integrated model of the electromechanical coupling effect between permanent-magnet synchronous motor excitation and electric vehicle transient dynamics, studied the mechanism of the deformation coupling loop of unbalanced electromagnetic forces in permanent-magnet synchronous motors, and proposed a multi-objective optimization method for an active-suspension system using a particle swarm optimization algorithm to address negative coupling effects. The optimized active suspension effectively reduces unbalanced electromagnetic forces caused by motor eccentricity and significantly improves electric-vehicle ride comfort. Tan [14] analyzed the influence of magnetic forces on the longitudinal- and lateral-coupling dynamics of vehicles and conducted extended research on different motor speeds. The results showed that magnetic forces have a certain negative impact on the longitudinal and lateral dynamics of vehicles, and must be considered for electric vehicles driven by hub motors. This study use a permanent-magnet synchronous motor (PMSM) as the vehicle’s propulsion motor and investigate the vertical motor excitation generated by the PMSM during the vehicle’s operation.

As an important component of a vehicle, the suspension system reduces the impact of uneven roads on the body and improves the vehicle’s ride comfort, handling stability, smoothness and safety. The traditional passive-suspension system has fixed parameters such as stiffness and damping, which are unable to effectively suppress vehicle vibrations under complex driving conditions. With the development of control theory and engineering, more and more researchers are exploring better control algorithms applied to active suspension to improve vehicle performance [15,16], including LQR control [17], optimal control algorithms [18], PID control algorithms [19], adaptive control algorithms [20], fuzzy control algorithms [21], genetic algorithms [22], neural network control algorithms [23], and so on. The traditional proportional–integral–derivative (PID) controller has been widely used due to its simple structure, good real-time performance, small computation, strong robustness, and low cost. However, since the control effect of traditional PID controllers depends directly on their proportional, derivative, and integral parameters, parameter tuning is a critical component of building a PID controller. Furthermore, since vehicle operating conditions and driving conditions are constantly changing, a suspension controller with high precision, high-performance processing capability, fast response, and stable reliability is required. Therefore, PID parameters should be combined with intelligent control methods for optimal performance. Pedro [24] proposed a nonlinear control method using dynamic neural network input–output feedback linearization. Particle swarm optimization was applied to train the dynamic neural network model and calculate the controller parameters. The effectiveness and robustness of the proposed controller were verified through time-domain and frequency-domain simulations. Han [25] focused on the uncertainty of vehicle suspension parameters and random road disturbances by using a flexible neural tree (FNT) to estimate the vertical disturbance of the road under uncertain vehicle parameters. The fitted-road power spectral density (PSD) was combined with real-time suspension performance to propose a road-estimated fuzzy proportional–integral–derivative (PID) control strategy. The results showed that the online fuzzy evaluation strategy could effectively reflect changes in road conditions, and the proposed adaptive fuzzy-PID-control strategy could adjust the parameters adaptively, according to road conditions, to meet control performance requirements. Kalaivani [26] proposed a new neural network-based vehicle active-suspension system (VASS) for vibration control under road disturbances. A proportional–integral–derivative (PID) controller was used to simulate VASS vibration control to train the neural network. The optimization results showed that the designed NN controller significantly reduced vehicle-body acceleration and ensured ride comfort.

Because it is difficult to describe the quantitative analysis between input and output, therefore, a BP neural network is chosen for learning. During the training process, the BP neural network can learn and automatically extract the implicit mapping relationship between input and output data, and can also adaptively memorize the learned content and store it in the network. Therefore, the BP neural network has high self-learning and adaptive abilities. However, because gradient descent is used in error backpropagation, it makes it easy for the solution to fall into the local optimum. In response to this issue, various swarm optimization algorithms have been developed to optimize BP neural networks. Intelligent particles in the swarm continuously learn historical data about themselves and the swarm to determine the parameter selection of the BP neural network. Currently, the most commonly used optimization algorithms for BP neural networks are particle swarm optimization (PSO) [27], grey wolf optimization (GWO) [28], the whale optimization algorithm (WOA) [29], and the sparrow search algorithm (SSA) [30]. These algorithms are inspired by the feeding behaviors of animals in nature. SSA is a recently proposed swarm-intelligence-optimization algorithm. Wang [31] established a fall-prediction model based on a BP neural network optimized by SSA, and conducted error prediction in fall detection. Experimental results showed that the improved BP neural network can avoid falling into local optimal solutions, achieving accurate fall detection. Xu [32] proposed a temperature control strategy based on SSA–Proportional–Integral–Derivative (PID) for a thermal system, which simulated current step changes, dynamic load changes of vehicles, and parameter changes in work. The results showed that this method has advantages such as fast convergence speed, good dynamic performance, and strong anti-interference ability. Huang [33] proposed a BP neural network adaptive-preview-control method for unmanned-vehicle path-tracking preview control; a preview time adjuster based on the SSA-BP neural network was established and a PID speed controller was designed to solve the influence of longitudinal speed changes on vehicle stability. The results showed that the proposed method has strong tracking ability at different speeds on different testing roads, and can be used during variable speed driving. Currently, there is relatively little research on suspension control utilizing swarm optimization algorithms. In comparison to other optimization algorithms, the sparrow optimization algorithm (SSA) stands out due to its adaptability, local search capabilities, and versatility. Furthermore, due to the simplicity of the algorithm, SSA lends itself well to parallelization and distributed implementation, which makes it potentially capable of solving large-scale optimization problems. Consequently, this study employed the sparrow optimization algorithm (SSA) to intelligently optimize neural networks, aiming to enhance vehicle ride comfort.

The above research scholars have provided us with many learning directions to explore the various fields of interaction between electric vehicles and road surfaces; however, there are still shortcomings. In terms of external stimuli on vehicles, a comprehensive consideration of the negative effects of vertical vibrations from hub motors requires multifaceted considerations. There is relatively limited research on compound stimuli under the coupling of motors and road surfaces. Furthermore, there is a lack of dynamic analysis of vehicles under various driving speeds and conditions.

In suspension control, swarm optimization algorithms have greater potential applications in semi-active-suspension control. The sparrow search algorithm (SSA), with its relatively new position in the field of intelligent optimization, holds promise for bringing new possibilities to the automotive domain based on its successful applications in other areas. Automotive systems often face multiple stimuli, such as different road conditions, driving speeds, and loads. The superiority of SSA lies in its ability to handle diverse search spaces, giving it an advantage in addressing scenarios with multiple stimuli. In the context of active-suspension vehicle-ride-comfort control, the current application of SSA is relatively limited. The well-established nature of PID controllers, their simplicity, and their proven effectiveness in a wide range of applications, including automotive systems, make them a suitable baseline for comparison. The choice of combining SSA optimization with BPNN neural network PID control not only fills gaps in this research area but also provides new perspectives and methods for the design of future control systems. By comparing the optimization-improvement effects of different swarm optimization algorithms under various vehicle operating conditions, it contributes to enhancing vehicle ride comfort and overall performance.

In the second section of this paper, the vertical dynamics relationship between hub-motor electric vehicles and road surfaces is considered. A vehicle dynamics model and various external-excitation input models are established. In the third section, an intelligent algorithm control strategy is constructed to establish a semi-active-suspension controller for the vehicle. In the simulation analysis of the fourth section, the actual effectiveness of the adopted control strategy is analyzed through a comparison of the results from simulated experiments. Finally, in the fifth section, a summary of the entire paper is provided.

## 2. Modeling of Electric Vehicle–Road-Surface Interaction

### 2.1. Dynamic Equations of Motion for Hub-Motor Electric Vehicle System

The vertical vibration of electric vehicles can be described using a two-degree-of-freedom quarter-vehicle model, while the road surface is modeled using a viscoelastic-foundation beam model. The road surface is simulated using Bernoulli–Euler beams of finite length supported at both ends, while the roadbed is simulated using a Kelvin viscoelastic foundation. The model is illustrated in Figure 1.

The Bernoulli–Euler beam model is based on the following assumptions:The elastic center of the vehicle coincides with its center of gravity;The vehicle body is rigid, and the movement of passengers is the same as that of the vehicle body;There is no sliding between the tires and the road surface, and the wheels always remain in contact with the ground;The vertical-vibration characteristics of the wheels are reduced by springs;Damping effects are not considered;It is assumed that the stiffness and damping of the vehicle suspension and tires are linear.

Based on D’Alembert’s principle and the principle of vibration, the equations for the vertical vibration of the vehicle and the road surface can be derived.
(1)$$\left.\begin{array}{c} m_{2}{\ddot{y}}_{2}-C_{2}({\dot{y}}_{1}-{\dot{y}}_{2})-K_{2}(y_{1}-y_{2})=F_{d}\\ (m_{1}+m_{3}){\ddot{y}}_{1}+C_{2}({\dot{y}}_{1}-{\dot{y}}_{2})+K_{1}(y_{1}-q-y_{r})+K_{2}(y_{1}-y_{2})=F(t)-F_{d} \end{array}\right\}$$
(2)$$\frac{\partial^{2}}{\partial x^{2}}\left(EI\frac{\partial^{2}y_{r}}{\partial x^{2}}\right)+Ky_{r}+C\frac{\partial y_{r}}{\partial t}+\rho A\frac{\partial^{2}y_{r}}{\partial t^{2}}=F\delta (x-vt)$$
where “*m*_1_” is the non-sprung mass; “*m*_2_” is the sprung mass; “ *m*_3_” is the mass of the wheel hub motor and the reduction mechanism; “*y*_1_” is the displacement of the non-sprung mass; “ *y*_2_” is the displacement of the sprung mass; “*y_r_*” is the secondary excitation of the road surface; “*q*” is the random excitation of the road surface; “*K*_1_” is the tire stiffness; “*K*_2_” is the suspension stiffness; “*F*_(*t*)_” is the vertical excitation of the motor; “*F_d_*” is the active control force acting on the suspension; “*K*” is the stiffness of the roadbed; “*C*” is the damping coefficient of the roadbed; “*C*_2_” is the damping coefficient of the suspension; “*F*” is the tire force; “*E*” is the elastic modulus of the road surface; “*I*” is the rotational inertia of the road surface; “*A*” is the cross-sectional area of the road surface; “*ρ*” is the density of the road surface; “*δ*” is the Dirac function; “*x*” is the displacement of the vehicle in the driving direction; and “*v*” is the driving speed of the vehicle.

### 2.2. Road-Excitation Model

#### 2.2.1. White-Noise Road Excitation

Different grades of road are usually expressed by the road roughness coefficient “*G_q_*”. According to the “Draft Method of Representing Road Unevenness” proposed by ISO/TC108/SC2N67, the power spectral density of the road can be represented as follows.
(3)$$G_{q}(n)=G_{q}(n_{0}){\left(\frac{n}{n_{0}}\right)}^{-w}$$
where “*n*” represents spatial frequency; “*n*_0_” represents the reference spatial frequency; *n*_0_ = 0.1 m^−1^, *G_q_*(*n*_0_) is the reference spatial frequency of road-power spectral density; and “*w*” is the frequency index, usually taken as *w* = 2.

The amplitude of road-surface roughness varies with vehicle speed, which is an important factor to consider when analyzing the dynamics of a vehicle’s suspension system. Convert the power spectral density of spatial frequency to *G_q_*(*n*) and the power spectral density of time frequency to *G_q_*(*f*). Vehicle variables can also be introduced, such as speed. When the vehicle travels on the road surface at a certain speed, the spatial frequency *n* and its equivalent time–frequency can be expressed as the following:(4)f=vn

In the equation, “*v*” represents the driving speed of the car in meters per second (m/s), and “*f*” represents the time–frequency in units of second, s^−1^.

Therefore, regarding the road-excitation model created using the filtered white-noise method, the following conclusions can be derived:(5)$$\dot{x}(t)+2\pi f_{0}x(t)=2\pi n_{0}\sqrt{G_{q}(n_{0})v}\omega (t)$$
where “*x*(*t*)” represents the road-surface displacement, “*ω*(*t*)” represents Gaussian white noise, “*f*_0_” represents the lower-cutoff spatial frequency, and “*v*” represents the driving speed of the car, *f*_0_ = 0.1 Hz.

This model is based on the concept of abstracting the random fluctuations of the road process as white noise that meets certain conditions. With an appropriate transformation system, a time-domain model of the road’s random roughness can be fitted. Roughness refers to the deviation between the road surface and an ideal plane, and the power-spectral-density geometric mean increases as the road grade decreases and the surface becomes rougher. We analyze and simulate C road surfaces with specific parameters, outlined in Table 1. The random C-level road surface is shown in Figure 2.

#### 2.2.2. Road-Surface Secondary Excitation

When a vehicle travels on a road, the unevenness of the road surface causes the vehicle to vibrate. The vehicle interacts with the road surface through tire forces, which in turn causes the road surface to vibrate and apply secondary excitation to the vehicle. The mathematical expression for road-surface vibration can be derived using the principle of modal superposition. The road surface is modeled as a viscoelastic beam on an elastic foundation, with a finite-length Bernoulli–Euler beam simulating the highway pavement, and a Kelvin viscoelastic foundation simulating the highway subgrade. The Bernoulli–Euler beam is subject to the following assumptions:The cross section of the beam is axisymmetric and much smaller in size than its length;The beam undergoes only planar motion, and the displacement is small;The influence of shear deformation is neglected;The stress along the thickness direction of the beam is zero;The effect of rotational inertia is neglected.

Boundary conditions of the beam:(6)$$y_{r}(x,t)\left|{\;}_{t=0}\right.=\frac{\partial y_{r}(x,t)}{\partial t}\left|{\;}_{t=0}\right.=0$$
(7)$$y_{r}(0)=y_{r}(L)=\frac{\partial^{2}y_{r}(0)}{\partial x^{2}}=\frac{\partial^{2}y_{r}(L)}{\partial x^{2}}=0$$

Using the assumed mode method, we can express the vertical displacement of the beam as the following:(8)$$y_{r}=\sum_{i=1}^{\infty }Y_{i}(x)\eta_{i}(t)$$

In the formula, the generalized coordinates *η_i_*(*t*) = *asin*(*ω_i_t + φ*), can be substituted into (2) to obtain the following:(9)$$EI\frac{\partial^{4}Y_{i}(x)}{\partial x^{4}}+(K-m\omega_{i}^{2})Y_{i}(x)=0$$

Regular mode functions can be derived from boundary conditions:(10)$$Y_{i}(x)=\sqrt{\frac{2}{mL}}\sin\frac{i\pi x}{L}$$

Using modal superposition can obtain the following:(11)$${\ddot{\eta }}_{i}(t)+2\xi_{i}\omega_{i}{\dot{\eta }}_{i}(t)+\omega_{i}^{2}\eta_{i}(t)=q_{i}(t)$$
where $\omega_{i}=\sqrt{{\left(\frac{i\pi }{L}\right)}^{4}\frac{EI}{m}+\frac{K}{m}}$, $\xi_{i}=\frac{C}{2m\omega_{i}}$, $\omega_{di}=\omega_{i}\sqrt{1-\xi^{2}}$.
(12)$$q_{i}(t)=\int_{0}^{L}F(t)\delta (x-x_{t})Y_{i}(x)dx=F(t)Y_{i}(x_{t})$$

According to the Duhamel integral, the road-surface response is obtained as follows:(13)$$y_{r}(x,t)=\sum_{i=0}^{\infty }\frac{Y_{i}(x)}{\omega_{di}}\int_{0}^{t}F(\tau )\sqrt{\frac{2}{mL}}\sin\left(\frac{i\pi }{2}+\frac{i\pi v\tau }{L}\right)\sin\left[\omega_{di}(t-\tau )\right]e^{-\xi_{i}\omega_{i}(t-\tau )}d\tau$$

Substituting $x=\frac{L}{2}+vt$ into the formula can obtain the secondary-displacement excitation of the road surface to the vehicle during driving [4,34]:(14)$$y_{r}(t)=\sum_{i=0}^{\infty }\frac{Y_{i}\left(\frac{L}{2}+vt\right)}{\omega_{di}}\int_{0}^{t}F(\tau )\sqrt{\frac{2}{mL}}\sin\left(\frac{i\pi }{2}+\frac{i\pi v\tau }{L}\right)\sin\left[\omega_{di}(t-\tau )\right]e^{-\xi_{i}\omega_{i}(t-\tau )}d\tau$$

The secondary-displacement excitation of the road surface is shown in Figure 3.

### 2.3. Motor Excitation

The disc-type permanent-magnet motor is a square-wave motor, with non-uniform air-gap magnetic density distribution leading to electromagnetic torque fluctuations. The tangential forces generated between the stator and rotor cause motor vibration, which exacerbates wheel vibration and affects vehicle smoothness.

The magnetic-field direction of the disc-type motor is axial and concentrated on the fan-shaped plane corresponding to the permanent magnet. The average electromotive force under a single magnetic pole can be obtained by integration over the fan-shaped plane:(15)$$E_{c}=\frac{1}{8}\omega \alpha_{i}B_{\delta }(D_{o}^{2}-D_{i}^{2})$$

In the formula, “*ω*” is the rotor angular velocity, *ω* = *πn*/30, “*α_i_*” is the calculation factor of the pole arc, “*B_δ_*” is the air-gap flux density amplitude, and “*D_o_*” and “*D*_i_” are the outer and inner diameters of the permanent magnet.

The electric loading at the average radius of the motor is the following:(16)$$A_{av}=\frac{4mNI_{\Phi }}{\pi (D_{o}+D_{i})}$$
where “*m*” is the number of winding turns, “*N*” is the number of winding turns connected in series per phase, and “*I*_Φ_” is the effective value of the current.

Collate (16) to obtain the following:(17)$$I_{\Phi }=\frac{A_{av}\pi (D_{o}+D_{i})}{4mN}$$

Electromagnetic power of the motor:(18)$$p=mE_{\Phi }I_{\Phi }=\frac{\pi }{2\sqrt{2}}mE_{\Phi av}I_{\Phi }$$

*E*_Φ*av*_ is the average value of each electromotive force, *E*_Φ*av*_ = 2*NEc.*

Substituting Equations (15) and (17) into Equation (18):(19)$$p=\frac{\pi^{2}}{32\sqrt{2}}\omega \alpha_{i}B_{\delta }A_{av}(D_{o}^{2}-D_{i}^{2})(D_{o}+D_{i})$$

From Formula (15) and Formula (19), the expression of the electromagnetic torque of the motor can be obtained as the following [35]:(20)$$T_{m}=p/\omega =\frac{\pi^{2}}{32\sqrt{2}}\alpha_{i}B_{\delta }A_{av}(D_{o}^{2}-D_{i}^{2})(D_{o}-D_{i})$$

The motor excitation is shown in Figure 4.

## 3. Construction of the Control Strategy

### 3.1. SSA-BPNN-PID Controller

When designing suspension-system control, four requirements need to be considered: road-holding stability, suspension dynamic deflection, and ride comfort: Ride comfort: ride comfort is closely related to vertical acceleration. Therefore, spring-loaded mass acceleration should be suppressed;Suspension displacement: if the suspension deformation is too large, it will collide with the limit block, affecting ride comfort. Therefore, the suspension deflection should be limited;Maintaining stability: in order to ensure vehicle safety and road-holding stability, tire hopping should be minimized;Maximum actuator force: since the power of the actuator is limited, the active control force provided by the suspension system should be limited by a threshold.

In the field of industrial automation control, PID is a simple and efficient control algorithm. PID control is used to generate errors and tracking between the actual output signal of the controlled object and the given signal. The control rate u of the system is obtained by proportional *P*, integral *I* and derivative *D* [36,37].

The neural network allows us to capture the nonlinearities and intricacies inherent in the system, providing a more adaptive and responsive control strategy. The optimization process, facilitated by the sparrow search algorithm, is employed to fine-tune the PID parameters within the neural network, ensuring that the control system achieves optimal performance under varying and challenging conditions.

The neural network consists of three layers: input, hidden, and output. In this article, a three-layer BP neural network is used as the controller, with three neurons in the input layer, five neurons in the hidden layer, and three neurons in the output layer. Each neuron is a PID neuron, consisting of a *P* neuron, an *I* neuron, and a *D* neuron, as shown in Figure 5. The neural network structure built in this article is 3-5-3.

The structure of the SSA-BPNN-PID controller for simulating a semi-active-suspension system is shown in Figure 6.

In Figure 6, the road excitation and motor excitation are inputs to the suspension control model, and the input error of the PID controller is represented by *e*(*k*); *u*(*k*) is the control quantity; the expected value of the system is represented by *r*(*k*), and the actual output value is represented by *y*(*k*). After optimizing the BP neural network algorithm through the sparrow algorithm, the best control parameters for the PID controller are obtained. Based on the optimal control parameters, the PID controller adjusts the size of the control quantity *u*(*k*) to achieve real-time control of the suspension system.

### 3.2. SSA Optimization Algorithm

The initial weight parameters for each layer of the BP neural network are usually selected as random numbers between −1 and 1. However, using random numbers as initial values can lead to slow convergence and greatly affect the training efficiency of the network, as well as causing the algorithm to converge to local extremes. Therefore, to obtain better training accuracy for the network, the sparrow search algorithm (SSA) in particle swarm optimization is used in this study to calculate and find the optimal parameters as the initial network weights for the BP neural network, to improve the efficiency of the network’s iterative convergence speed and to find optimal control.

The sparrow population in SSA is divided into two parts: producers and predators. Producers have high adaptability and energy reserves, and their main task is to provide directions and areas for predators to search for food. The search range of producers is larger than that of predators. Predators follow producers to find food and obtain their energy reserves, thereby increasing their adaptability. Some predators increase their energy reserves by predation and thus become producers. In addition, some sparrows in the population serve as early warning systems. When danger approaches, the early warning system sends out a warning signal and spreads to the safe area to obtain a better position. When the alarm value exceeds the set threshold, the producers will lead all the predators to leave the danger zone. SSA has better global-search and local-development capabilities and can consider all variable factors of the population, enabling the population to quickly enter the optimal position. SSA also has the advantages of fewer iterations and higher prediction-model accuracy [38,39]. In this study, the sparrow search algorithm is used to optimize the weights and thresholds of the BP neural network to obtain more accurate results.

The position of a sparrow can be represented by the following matrix:(21)$$X=\left[\begin{array}{cccc} x_{1,1} & x_{1,2} & \cdots & x_{1,d}\\ x_{2,1} & x_{2,2} & \cdots & x_{2,d}\\ \vdots & \vdots & \vdots & \vdots \\ x_{n,1} & x_{n,2} & \cdots & x_{n,d} \end{array}\right]$$

Here, “*n*” is the number of sparrows, and “*d*” represents the dimensionality of the variables to be optimized. The sparrow population is initialized and fitness is calculated to obtain individual fitness values, global optimum, worst fitness value, and their corresponding positions. The fitness values of all sparrows are as follows:(22)$$F_{X}=\left[\begin{array}{cccc} f([x_{1,1} & x_{1,2} & \cdots & x_{1,d}]\\ f([x_{2,1} & x_{2,2} & \cdots & x_{2,d}]\\ \vdots & \vdots & \vdots & \vdots \\ f([x_{n,1} & x_{n,2} & \cdots & x_{n,d}] \end{array}\right]$$

The values in each row of “*F_x_*“ represent the fitness values of individuals. In SSA, producers with better fitness values have priority in obtaining food during the search process.

Moreover, since producers are responsible for searching for food and guiding the movement of the entire population, they can search for food in a wider area than predators. During each iteration, the positions of producers are updated as follows:(23)$$X_{i,j}^{t+1}=\left\{\begin{array}{ll} X_{i,j}^{t}\cdot \exp\left(\frac{-i}{\alpha \times {\mathrm{iter}}_{\max}}\right) & R_{2}<ST\\ X_{i,j}^{t}+Q\cdot L & R_{2}\geq ST \end{array}\right.$$
where “*t”* denotes the current iteration number, “*i*” is the number of sparrows, “*j*” is the dimensionality of the optimization problem, representing the position information of the *i*-th sparrow in the *j*-th dimension at the *t*-th optimization, and “*itermax*” denotes the maximum number of iterations. *A*∈(0, 1] is a random number, *R_2_*∈(1, 0) represents the warning value, and *ST*∈[0.5, 1] represents the safety value −0.8.

If *R*_2_ < *ST*, the population is considered safe and can explore a wide area in search of food. If *R*_2_ ≥ *ST*, it means there are predators in the vicinity, and the entire population is alerted to abandon their search and immediately fly to a safe area. *Q* is a random number that follows a normal distribution, and *L* is a 1 × d matrix with all elements equal to 1.

Some species with less favorable foraging locations are more likely to fly to other areas in search of additional sources of food. The updated descriptions of the locations are as follows:(24)$$X_{i,j}^{t+1}=\left\{\begin{array}{ll} Q\cdot \exp\left(\frac{X_{worst}-X_{i,j}^{t}}{i^{2}}\right) & i>\frac{n}{2}\\ X_{i,j}^{t+1}+\left|X_{i,j}^{t}-X_{p}^{t+1}\right|\cdot A^{+}\cdot L & i\leq \frac{n}{2} \end{array}\right.$$
where “*X_p_*” is the best position currently occupied by the discoverer, “*X_worst_*” represents the current global worst position, and “*A*” is a 1 × *d* matrix, with each element being 1 or −1, *A^+^* = *A^T^*(*AA^T^*). *i* > *n*/2 indicates that the joiner with a lower fitness value has not obtained food, is starving, and needs to fly to other places to find more food.

Ten percent of the individuals are selected for vigilance work, and the entire population is anti-looting when facing danger. The position update of the security police is as follows:(25)$$X_{i,j}^{t+1}=\left\{\begin{array}{ll} X_{best}^{t}+\tau \left|X_{i,j}^{t}-X_{best}^{t}\right| & \;\;f_{i}>f_{g}\\ X_{i,j}^{t}+H\left[\frac{\left|X_{i,j}^{t}-X_{worst}^{t}\right|}{\left(f_{i}-f_{w}\right)+\varepsilon }\right] & \;\;f_{i}=f_{g} \end{array}\right.$$
where “*X_best_*” is the current global best position, “*τ*” is a normally distributed random number in the range of 0 to 1 used as a control step-size parameter, “*H*”∈[−1, 1] is a random number, “*f_i_*”, “*f_g_*”, and “*f_w_*” denote the current individual sparrow fitness value, and the current global best and worst fitness values, respectively, and “*ε*” is a constant to prevent the denominator from being zero. When *f_i_* = *f_g_*, the individual at the middle position detects danger and updates its position; when *f_i_* > *f_g_*, individuals at the edges are alerted and update their positions.

The flowchart of SSA-BP-PID is shown in Figure 7.

## 4. Simulation and Analysis

### 4.1. Quarter-Car Model Parameters and Driving Conditions

The model was simulated using MATLAB/Simulink (2022A), and the simulation model of 1/4 passive-suspension and active-suspension systems was established. Table 2 shows the basic parameters of the model.

To highlight the optimization effects of the SSA-BPNN-PID control strategy and validate its effectiveness, simulations, and analyses were conducted on passive suspension, PID, PSO-BPNN-PID, and SSA-BPNN-PID active suspensions, respectively.

The SSA-BP-PID control algorithm uses SSA to control the BP neural network. The threshold and weights are optimized, and each individual calculates its fitness value through an adaptive function and a series of update operations. The BPNN algorithm continuously updates the weights of the network based on the rates of these optimal individuals. The control output provides the best control parameters. The parameter settings of the SSA prediction model are shown in Table 3.

Select an appropriate fitness function from the suspension controller and integrate the three sub objective functions into a single objective function. The selected suspension-system fitness function, where ${BA}_{a}$, ${SD}_{a}$, and ${DTL}_{a}$ are the root mean square values of the vehicle-smoothness-evaluation indicators under the active-suspension control strategy, and ${BA}_{p}$, ${SD}_{p}$, and ${DTL}_{p}$ are the root mean square values of the vehicle-smoothness-evaluation indicators under the passive dynamic-suspension control strategy. The fitness equation is as follows:(26)$$\min f(x)=\frac{BA_{a}}{BA_{p}}+\frac{SD_{a}}{SD_{p}}+\frac{DTL_{a}}{DTL_{p}}$$

### 4.2. Comparison between PID Control and SSA-BPNN-PID Control

This study focused on passenger vehicles in motion and investigated low-speed conditions (30 km/h, 40 km/h, 50 km/h), medium-speed conditions (60 km/h, 70 km/h, 80 km/h), and high-speed conditions (90 km/h, 100 km/h, 110 km/h) on a C-grade road surface.

In the optimization of the passive-suspension system, PID control and SSA-BPNN-PID control were utilized. The optimized simulation results were compared, and the impact of the control strategies on the improvement of the suspension system was analyzed through three ride-comfort evaluation metrics: body acceleration, suspension deflection, and dynamic tire load. By establishing tables, the changes in the root mean square values of performance indicators under different operating conditions were compared.

Under the C-grade random road, a comparison was made between passive suspension and active-control suspension, PID control, and SSA-BPNN-PID control based on time-domain simulation results at different vehicle speeds, as shown in Figure 8, Figure 9 and Figure 10. From the figures, it is evident that the optimization effects of PID control and SSA-BPNN-PID control are prominently visible. Compared to the passive-suspension system, significant disparities in the time-domain curves of optimized body acceleration, suspension deflection, and dynamic tire load are apparent across various operating conditions. The adoption of PID control and SSA-BPNN-PID control effectively reduces body acceleration and suspension deflection, with the magnitude of reduction increasing as vehicle speed conditions rise. Furthermore, PID control and SSA-BPNN-PID control lead to an increase in dynamic tire load, with the magnitude of the increase also growing in tandem with higher vehicle-speed conditions.

To provide a more intuitive analysis of the control effectiveness of different controllers on the suspension system, data processing was performed on the aforementioned charts to obtain the root mean square (RMS) values for each curve, as shown in Table 4, Table 5 and Table 6. Body acceleration is a primary parameter affecting ride comfort; therefore, minimizing overall vehicle-body acceleration to the greatest extent is crucial. PID control can effectively reduce body acceleration, and the optimization effect of SSA-BPNN-PID control is superior to PID control. Under various vehicle driving conditions, the RMS value optimized by SSA-BPNN-PID control is approximately 11% to 15% lower. Moreover, as the vehicle speed increases, the reduction becomes more significant. In the case of suspension deflection, the proportion of RMS reduction is even greater, ranging from approximately 10% to 21%. Typically, as vehicle speed increases, the reduction becomes more pronounced. Overall, SSA-BPNN-PID control exhibited lower RMS values for both body acceleration and suspension deflection, indicating better optimization performance. Regarding dynamic tire load, the RMS values of the optimized dynamic tire load under PID control and SSA-BPNN-PID control show a slight increase compared to the passive suspension, with an increase ranging from 3% to 17%. This increase in dynamic tire load falls within a reasonable range. Sacrificing a certain level of dynamic tire load to reduce body acceleration and suspension deflection is reasonable within a certain range. Therefore, it can be concluded that both PID and SSA-BPNN-PID active-suspension control strategies can enhance vehicle ride comfort. Furthermore, as vehicle speed increases, the reduction in body acceleration and suspension deflection becomes more pronounced, and the increase in dynamic tire load becomes more noticeable. Additionally, the SSA-BPNN-PID active-suspension control strategy outperforms the PID active-suspension control strategy.

### 4.3. Comparison of Suspension Control between PSO-BPNN-PID and SSA-BPNN-PID

The swarm-intelligence-optimization method, Particle Swarm Optimization (PSO), is often employed for multi-objective optimization situations [40,41]. A comparison was made between the PSO-BPNN-PID optimization control strategy and the SSA-BPNN-PID optimization control strategy, and the simulation results are shown in the following figures.

Similarly, low-speed, medium-speed, and high-speed conditions on C-class road surfaces were studied by comparison. PSO-BPNN-PID control and SSA-BPNN-PID control were employed. The optimized simulation results were compared, and the impact of the control strategies on the improvement of the suspension system was analyzed.

Similarly, under C-grade random road conditions, a comparison was conducted between PSO-BPNN-PID control and SSA-BPNN-PID control based on time-domain simulation results at different vehicle speeds. Figure 11, Figure 12 and Figure 13 illustrate that the adoption of both PSO-BPNN-PID control and SSA-BPNN-PID control effectively reduces body acceleration and suspension deflection. Significant disparities in body acceleration, suspension deflection, and dynamic tire load are apparent across various operating conditions, with the magnitude of reduction increasing as vehicle speed conditions rise. Concurrently, both PSO-BPNN-PID control and SSA-BPNN-PID control lead to an increase in dynamic tire load.

As shown in Table 7, Table 8 and Table 9, in terms of body acceleration, the RMS values showed a reduction ranging from approximately 0% to 5% under various driving conditions. Regarding suspension deflection, the reduction ranged from 1% to 11%. SSA-BPNN-PID control exhibited lower RMS values for body acceleration and suspension deflection, while the RMS value for dynamic tire load under SSA-BPNN-PID control, although showing a slight increase compared to PSO-BPNN-PID control (in the range of 1% to 12%), remained within a reasonable range. It can be concluded that the optimization effects of SSA-BPNN-PID control are superior to PSO-BPNN-PID control, indicating that SSA-BPNN-PID control effectively enhances vehicle ride comfort.

## 5. Discussion and Conclusions

To improve the ride comfort and smoothness of hub-motor electric vehicles, this study first established a random road surface. Building upon the excitation provided by the random road surface, it considered the secondary excitation caused by the interaction between the hub motor and the road surface. Subsequently, it investigated the motor excitation within the hub motor, studying the vertical semi-active-suspension vehicle–road-coupling dynamics model under multi-source excitation and designed the SSA-BPNN-PID control optimization algorithm.

Simulation analysis indicated that, in terms of body acceleration, compared to passive suspension, the adoption of the SSA-BPNN-PID optimization control algorithm resulted in a reduction of 10% to 16%. Suspension deflection decreased by 10% to 21% with the SSA-BPNN-PID optimization control algorithm, and the dynamic tire load increased by 3% to 17% under SSA-BPNN-PID optimization control. The actively controlled suspension system with SSA-BPNN-PID control exhibited superior control performance compared to the passive-suspension system with PID control and the actively controlled suspension system with PSO-BPNN-PID control. It effectively attenuated vehicle body vibrations induced by external disturbances.

Graphical and data analysis demonstrated that both PSO-BPNN-PID and SSA-BPNN-PID controllers provided improved vibration control compared to the passive system. However, the BPNN-PID controller based on the sparrow search algorithm (SSA) outperformed the others in terms of vibration control. In body acceleration, it achieved a reduction of up to 1% to 6%, suspension deflection showed a reduction of 1% to 10%, and dynamic tire load increased by 1% to 11% more than in the PSO-BPNN-PID controller. In comparison, the SSA-optimized BPNN-PID controller exhibited significantly superior performance.

This paper focused on addressing the negative effects of vertical vibrations in hub-motor electric vehicles. We proposed a semi-active-suspension control approach for vehicles considering compound stimuli, and optimized the BP neural network PID controller using the sparrow search algorithm. Based on considerations of various driving conditions, the proposed semi-active-suspension controller demonstrated significant improvements in enhancing vehicle ride comfort when compared with other controllers.

## Figures and Tables

**Figure 1 sensors-24-01757-f001:**
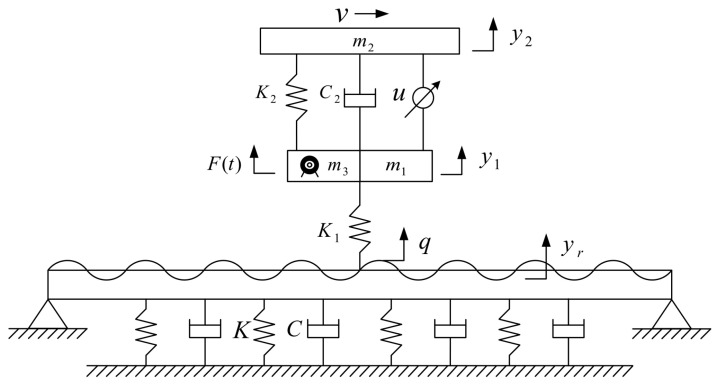
Dynamic model of electric vehicle wheel-hub motor system coupled with road.

**Figure 2 sensors-24-01757-f002:**
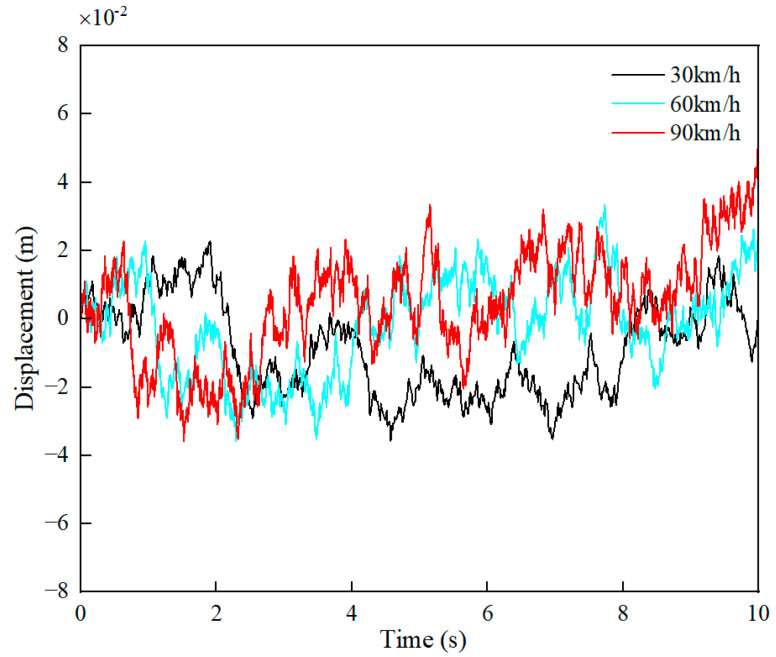
Random excitation of C-level road surface.

**Figure 3 sensors-24-01757-f003:**
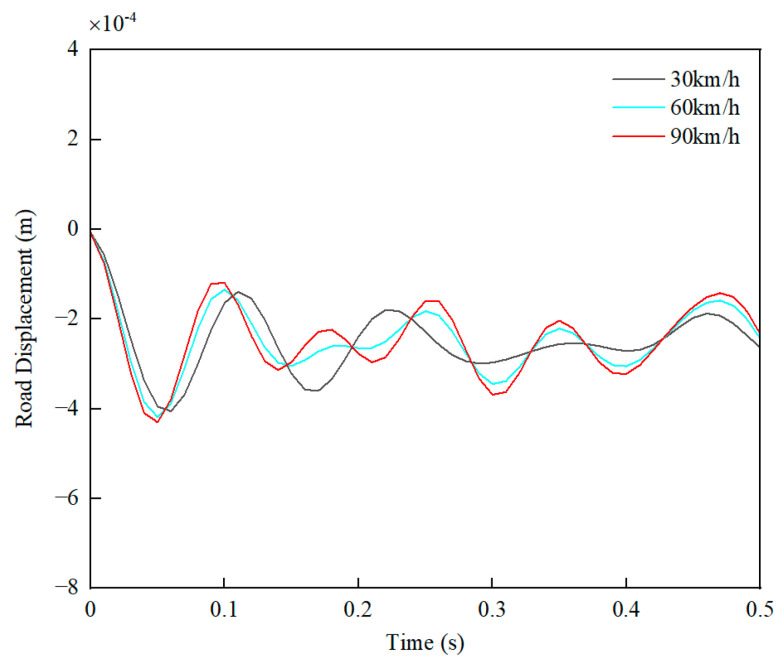
The displacement of the road surface due to the secondary excitation.

**Figure 4 sensors-24-01757-f004:**
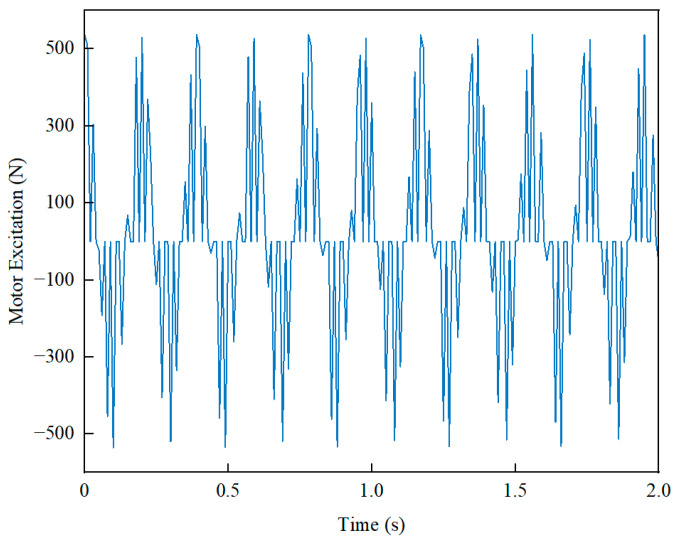
Disk-motor vertical excitation.

**Figure 5 sensors-24-01757-f005:**
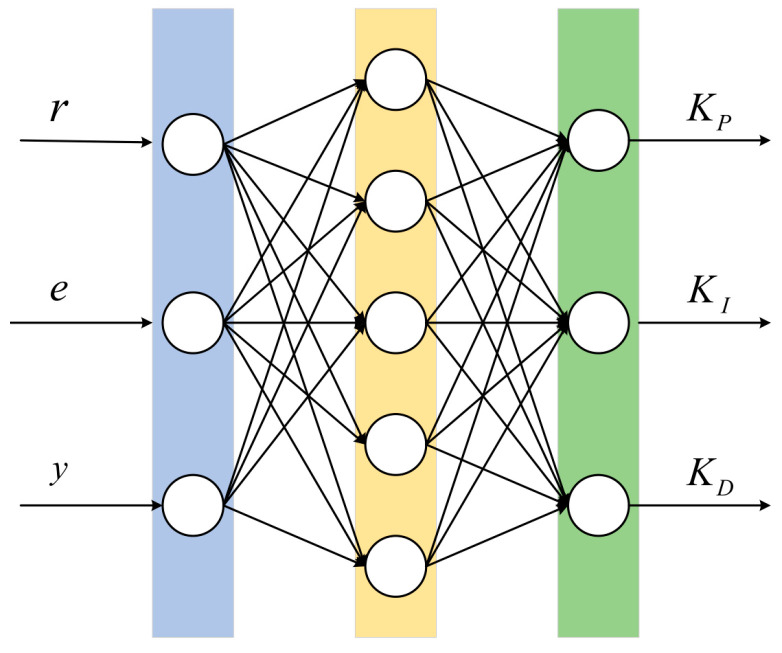
Structure of back-propagation neural network.

**Figure 6 sensors-24-01757-f006:**
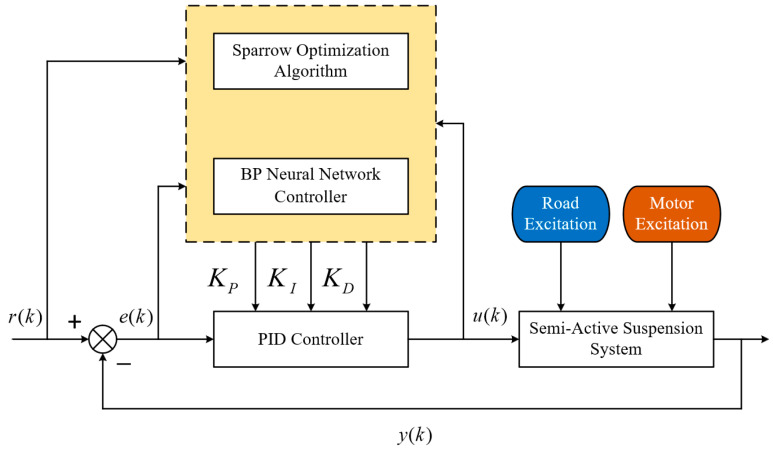
Structure of SSA-BPNN-PID controller.

**Figure 7 sensors-24-01757-f007:**
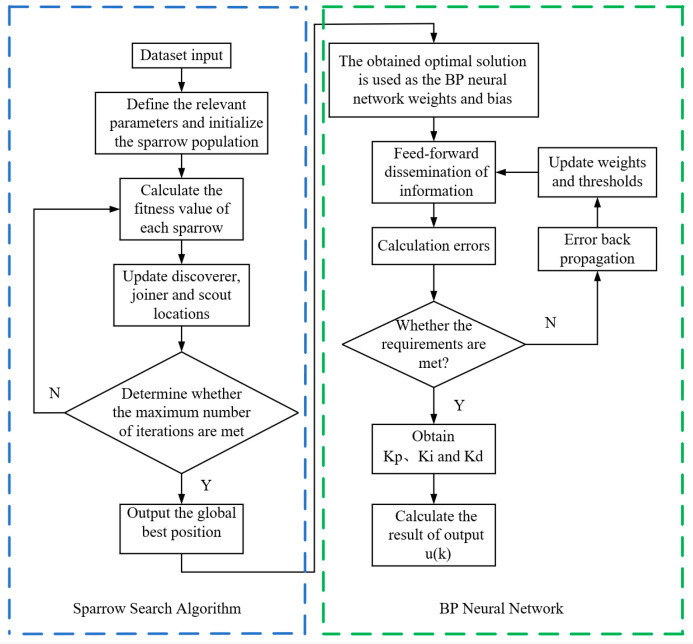
SSA-BP-PID flowchart.

**Figure 8 sensors-24-01757-f008:**
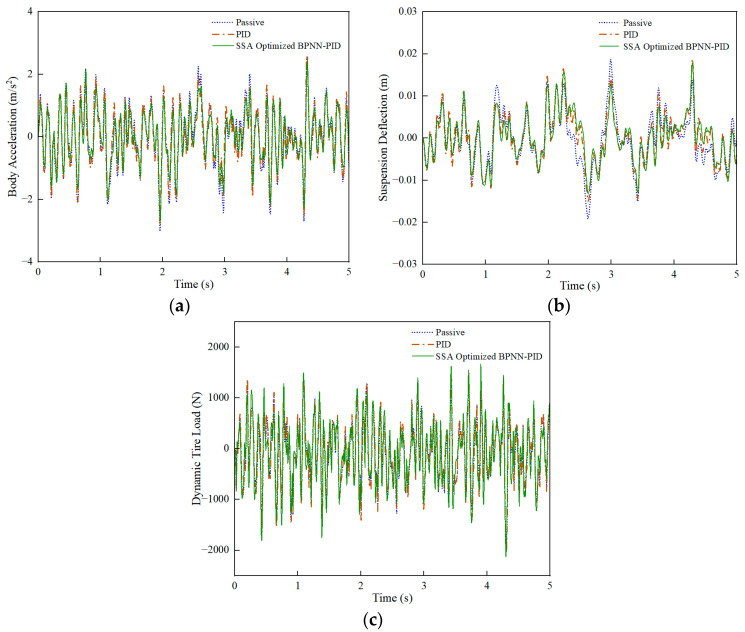
30 km/h-vehicle driving conditions. (**a**) Body acceleration, (**b**) suspension deflection, (**c**) dynamic tire load.

**Figure 9 sensors-24-01757-f009:**
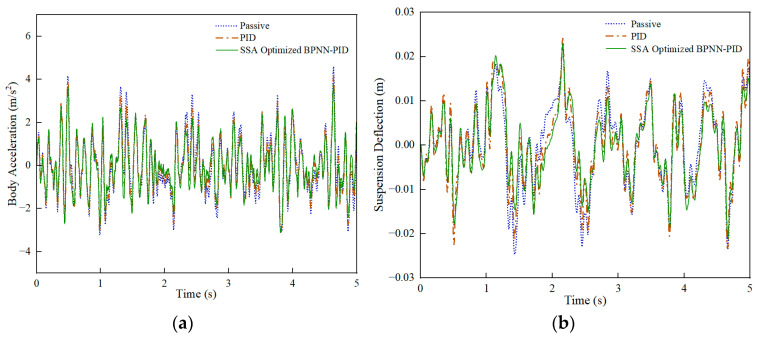
60 km/h-vehicle driving conditions. (**a**) Body acceleration, (**b**) suspension deflection, (**c**) dynamic tire load.

**Figure 10 sensors-24-01757-f010:**
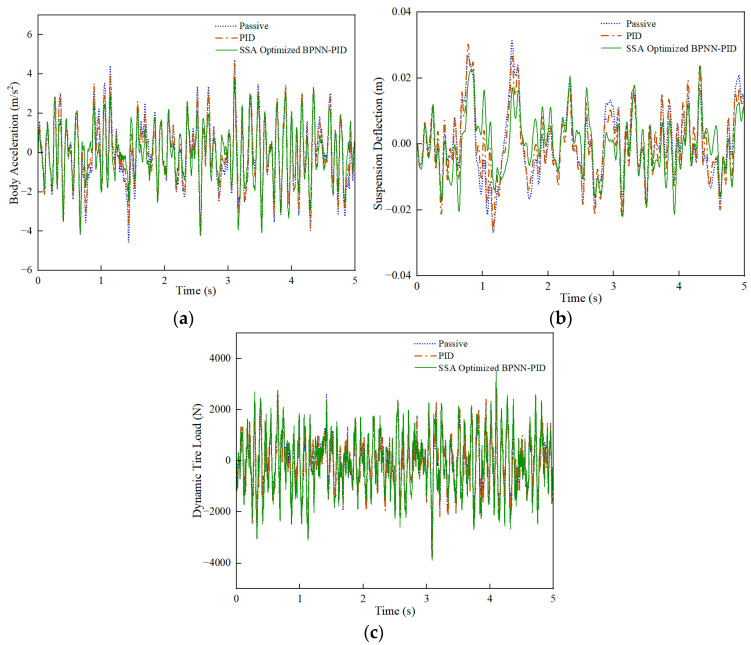
90 km/h-vehicle driving conditions. (**a**) Body acceleration, (**b**) suspension deflection, (**c**) dynamic tire load.

**Figure 11 sensors-24-01757-f011:**
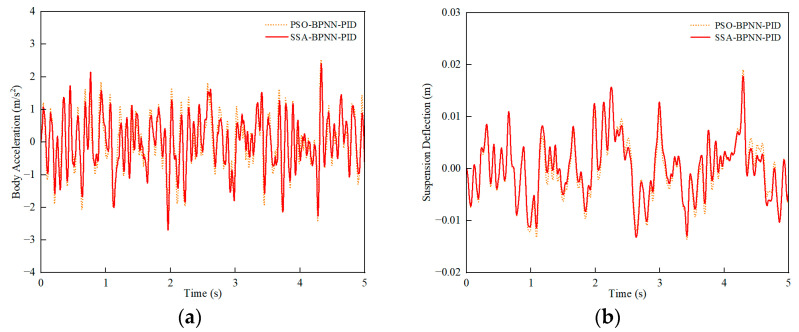
30 km/h-vehicle driving conditions. (**a**) Body acceleration, (**b**) suspension deflection, (**c**) dynamic tire load.

**Figure 12 sensors-24-01757-f012:**
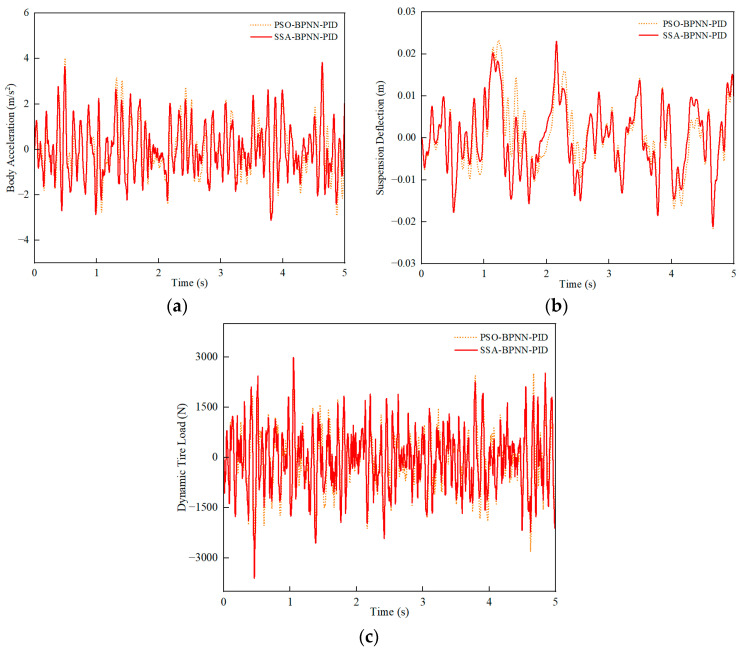
60-km/h vehicle driving conditions. (**a**) Body acceleration, (**b**) suspension deflection, (**c**) dynamic tire load.

**Figure 13 sensors-24-01757-f013:**
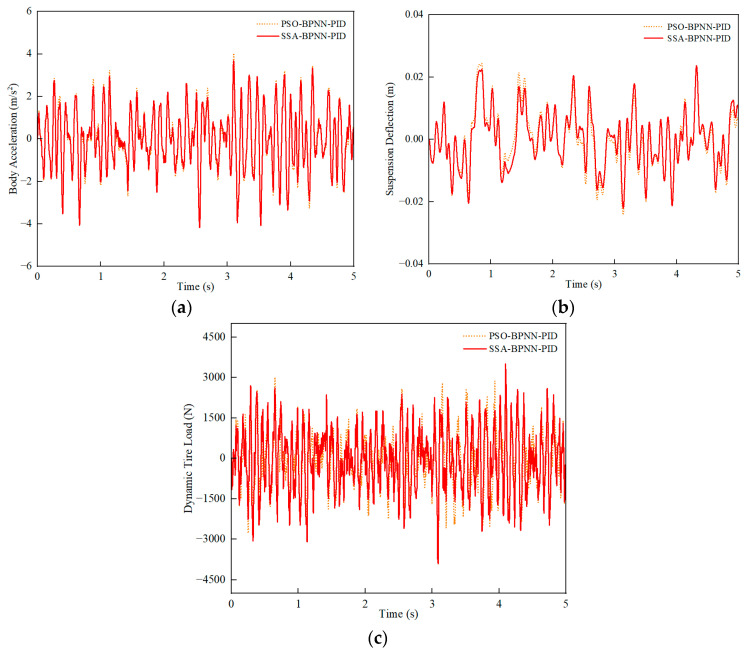
90 km/h-vehicle driving conditions. (**a**) Body acceleration, (**b**) suspension deflection, (**c**) dynamic tire load.

**Table 1 sensors-24-01757-t001:** Parameters of Various Grades of Pavement.

Road-Surface Grade	Geometric Mean of Power Spectral Density *G_q_*(*n*_0_)/10^−6^m^3^
A	16
B	64
C	256

**Table 2 sensors-24-01757-t002:** Basic parameters of the model T.

Item	Notation	Value
Vehicle		
Tire mass	*m* _1_	35 kg
Vehicle-body mass	*m* _2_	310 kg
Tire stiffness	*K* _1_	2 × 10^5^ N/m
Suspension stiffness	*K* _2_	1.96 × 10^4^ N/m
Suspension damping	*C* _2_	1695 N·s/m
Bub Motor		
Hub-motor mass	*m_d_*	20 kg
Polar logarithm	*α_i_*	2/π
Number of winding items	*m*	3
Motor speed	*n*	231~693 r/min
Air-gap flux-density amplitude	*B_δ_*	0.7
Winding turns per phase of a stator winding	*N*	208
Rated current	*I* _Φ_	12.85 A
Permanent-magnet outer diameter	*D_o_*	0.32 m
Permanent-magnet inner diameter	*D_i_*	0.185 m
Foundation and Beam		
Road length	*L*	140 m
Road width	*b*	6 m
Road thickness	*h*	0.1 m
Pavement modulus of elasticity	*E*	1.6 × 10^9^ N/m^2^
Concrete density	*ρ*	2.5 × 10^3^ kg/m^3^
Subgrade stiffness	*K*	8 × 10^6^ N/m^2^
Subgrade damping coefficient	*C*	3 × 10^5^ N·s/m^2^

**Table 3 sensors-24-01757-t003:** Parameter Settings of the SSA Prediction Model.

SSA	Parameter Setting
Maximum number of iterations	200
Initial population size	50
Percentage of discoverers	0.2
Proportion of alerts	0.2
Safety value	0.8

**Table 4 sensors-24-01757-t004:** Comparison results of Root Mean Square (RMS) Values for Passive Suspension, PID Optimized Strategy, and SSA Optimized BPNN-PID Strategy under Low-Speed Driving Conditions.

Evaluation Indices	Velocity (km/h)	Passive	PID	SSA-BPNN-PID	Reduction	
		RMS	RMS	RMS	Compare with Passive	Compare with PID
Body Acceleration (m/s^2^)	30	0.9420	0.8818	0.8113	−6.39%	−8.00%
	40	1.0857	1.0112	0.9561	−6.86%	−5.45%
	50	1.2544	1.1569	1.0856	−7.77%	−6.16%
Suspension Deflection (m)	30	0.006341	0.005995	0.005682	−5.46%	−5.22%
	40	0.007322	0.006959	0.005724	−4.96%	−17.75%
	50	0.008890	0.008330	0.007760	−6.30%	−6.84%
Dynamic Tire Load (N)	30	578.38	593.54	618.61	2.62%	4.22%
	40	670.66	687.58	696.97	2.52%	1.37%
	50	773.16	788.15	848.15	1.94%	7.61%

**Table 5 sensors-24-01757-t005:** Comparison results of Root Mean Square (RMS) Values for Passive Suspension, PID Optimized Strategy, and SSA Optimized BPNN-PID Strategy under Medium-Speed Driving Conditions.

Evaluation Indices	Velocity (km/h)	Passive	PID	SSA-BPNN-PID	Reduction	
		RMS	RMS	RMS	Compare with Passive	Compare with PID
Body Acceleration (m/s^2^)	60	1.3577	1.2499	1.1690	−7.94%	−6.47%
	70	1.4535	1.3364	1.2431	−8.06%	−6.98%
	80	1.5670	1.4561	1.3282	−7.08%	−8.78%
Suspension Deflection (m)	60	0.009685	0.009070	0.008271	−6.35%	−8.81%
	70	0.010408	0.009745	0.008877	−6.37%	−8.91%
	80	0.010190	0.009762	0.008251	−4.20%	−15.48%
Dynamic Tire Load (N)	60	833.65	848.95	916.76	1.84%	7.99%
	70	890.31	905.98	980.83	1.76%	8.26%
	80	918.92	935.23	1077.70	1.77%	15.23%

**Table 6 sensors-24-01757-t006:** Comparison results of Root Mean Square (RMS) Values for Passive Suspension, PID Optimized Strategy, and SSA Optimized BPNN-PID Strategy under Medium-Speed Driving Conditions.

Evaluation Indices	Velocity (km/h)	Passive	PID	SSA-BPNN-PID	Reduction	
		RMS	RMS	RMS	Compare with Passive	Compare with PID
Body Acceleration (m/s^2^)	90	1.6513	1.5347	1.3931	−7.06%	−9.23%
	100	1.7302	1.6083	1.4613	−7.05%	−9.14%
	110	1.8051	1.6784	1.5336	−7.02%	−8.63%
Suspension Deflection (m)	90	0.010763	0.010297	0.008963	−4.33%	−12.96%
	100	0.011297	0.010796	0.009167	−4.43%	−15.09%
	110	0.011795	0.011265	0.009632	−4.49%	−14.50%
Dynamic Tire Load (N)	90	970.63	988.22	1132.56	1.81%	14.61%
	100	1019.25	1038.01	1184.45	1.84%	14.11%
	110	1065.55	1085.44	1234.05	1.87%	13.69%

**Table 7 sensors-24-01757-t007:** Comparison results of Root Mean Square (RMS) Values for PSO Optimized BPNN-PID Strategy and SSA Optimized BPNN-PID Strategy under Low-Speed Driving Conditions.

Evaluation Indices	Velocity (km/h)	PSO-BPNN-PID	SSA-BPNN-PID	Reduction
		RMS	RMS	
Body Acceleration (m/s^2^)	30	0.8543	0.8113	−5.03%
	40	1.0049	0.9561	−4.86%
	50	1.0863	1.0856	−0.06%
Suspension Deflection (m)	30	0.005807	0.005682	−2.15%
	40	0.006406	0.005724	−10.65%
	50	0.007607	0.007760	2.01%
Dynamic Tire Load (N)	30	605.22	618.61	2.21%
	40	693.38	696.97	0.52%
	50	829.01	848.15	2.31%

**Table 8 sensors-24-01757-t008:** Comparison results of Root Mean Square (RMS) Values for PSO Optimized BPNN-PID Strategy and SSA Optimized BPNN-PID Strategy under Low-Speed Driving Conditions.

Evaluation Indices	Velocity (km/h)	PSO-BPNN-PID	SSA-BPNN-PID	Reduction
		RMS	RMS	
Body Acceleration (m/s^2^)	60	1.1786	1.1690	−0.81%
	70	1.2640	1.2431	−1.65%
	80	1.4110	1.3282	−5.87%
Suspension Deflection (m)	60	0.008422	0.008271	−1.79%
	70	0.009068	0.008877	−2.11%
	80	0.008412	0.008251	−1.91%
Dynamic Tire Load (N)	60	888.92	916.76	3.13%
	70	948.19	980.83	3.44%
	80	968.26	1077.70	11.30%

**Table 9 sensors-24-01757-t009:** Comparison results of Root Mean Square (RMS) Values for PSO Optimized BPNN-PID Strategy and SSA Optimized BPNN-PID Strategy under Low-Speed Driving Conditions.

Evaluation Indices	Velocity (km/h)	PSO-BPNN-PID	SSA-BPNN-PID	Reduction
		RMS	RMS	
Body Acceleration (m/s^2^)	90	1.4191	1.3931	−1.83%
	100	1.5406	1.4613	−5.15%
	110	1.5856	1.5336	−3.28%
Suspension Deflection (m)	90	0.009176	0.008963	−2.32%
	100	0.009543	0.009167	−3.94%
	110	0.009980	0.009633	−3.48%
Dynamic Tire Load (N)	90	1066.95	1132.56	6.15%
	100	1098.77	1184.45	7.80%
	110	1186.75	1234.05	3.99%

## Data Availability

Data are contained within the article.

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
