# Peer review of "Optimization of the Semi-Active-Suspension Control of BP Neural Network PID Based on the Sparrow Search Algorithm"

_sensors, 2024, doi:10.3390/s24061757_

Round 1
Reviewer 1 Report
Comments and Suggestions for Authors
The following major comments need to be addressed:
1. The paper lacks major novelty. There is no novelty in PID, BPNN and SSA algorithms.
2. The comparison of SSA with other optimization techniques is missing.
3. Add the pseudocode of SSA.
4. The control algorithm for the considered system is very well proved in the literature. Thus, the motive behind authors work is missing.
5. Why PID? various advanced or modified PID are proposed in the literature.
6. For designing a simple controllers with only 3 parameters why neural network and also an optimization?
7. No performance analysis for external disturbance
8. No comparison of techniques in the literature
Comments on the Quality of English Language
Fine
Reviewer 2 Report
Comments and Suggestions for Authors
This work presents a study on the vertical dynamic coupling effect model of disc-type permanent magnet motors and hub motors in electric vehicles, which contribute to increased unsprung mass and intensified vibration. A coupled dynamics model of a semi-active suspension vehicle-road system is investigated under multiple excitations, considering motor excitation and vehicle-road coupling. A BP neural network PID controller optimized using the Sparrow Search Algorithm is proposed, demonstrating improved ride comfort for hub motor electric vehicles compared to conventional PID control and PSO. The following comments may be of help to improve its completeness and eligibility for publication.
1) Although the authors present the distinctive characteristics and justifications of this work throughout the introduction, I think it would be beneficial to add a paragraph at the end of the introduction highlighting the novelties and objectives of this study.
2) The graphic quality of the all figures used in the manuscript could be improved and uniformed, since some of them are difficult to read (text size, subtitles, legends, grids, and resolution).
3) Is Figure 2 correct? The random excitation of the road surface (q) varies from -4 m to 4 m in a time interval of 10 seconds? I believe there may be an error in the unit of measurement of the graph.
4) Chapter 3.1 line 309, the authors state: "… three requirements need to be considered:". However, just below, they present 4 requirements.
5) Line 324, rewriting the paragraph, is very confusing. I suggest starting the paragraph with the sentence: “The neural network consists of three layers: input hidden, and output” (line 325).
6) Present and justify in the text the type of activation function used in the neurons of the artificial neural network. Is there a reason for choosing a hidden layer with five neurons?
7) Present in the text how the fitness value is calculated (equation).
8) Enhance the text to highlight the conditions under which Figures 8-10 and Tables 4-6 were evaluated.
Comments on the Quality of English Language
The quality of the English language is good, I didn't find any problems during my reading.
Reviewer 3 Report
Comments and Suggestions for Authors
The paper addresses an interesting topic in the field of automatic control systems, namely, the development of a semi-active suspension system for vibrations mitigation in electric vehicles. It is shown that the authors semi-active suspension improves ride comfort compared to PSO-BPNN-PID (PID control and Particle Swarm Optimization) and meets control performance requirements under different driving conditions. Numerical simulations are used to discuss the effectiveness of the proposed solution. Comments on the paper are hereafter.
The introduction should end with a clear and concise description of the organization of the paper, in such a way that (since the beginning) the reader have a clear picture about the content of the sections, e.g.: in Section 2 “Generative adversarial network model based on scene constraints…” is introduced; in Section 3 “Experimental results…” are reported; etc. etc.
Moreover, it is a general practice to put the cited articles right after the author’s name and not after several lines, e.g.: line 42, “Hu studied the…”, should be “Hu [4] studied the…”. Similarly: line 49 should be “Zheng [5] investigated…”. The same at line 55 “Snehasagar [6]…”, line 60 “Li [7]…”, line 66 “Krishnanunni [9]…”. Please check also the subsequent lines.
Also check minor typos within the entire document, e.g.: line 13, “consider” should be “considering”; line 30, “driven” should be “driving”; line 58, “…equation, the…” should be “…equation. The…”; etc. etc.
Some sentences are a bit convoluted (although one may get the meaning), e.g., “The vertical dynamic coupling effect model of disc-type permanent magnet motors and hub motors is proposed…”. The entire sentence appears as the subject of the verb. Please consider rewriting sentences like this and check the whole document to simplify the reading.
The caption of “Figure 4. Motor excitation.” (and other short captions) could be improved. Please consider providing further information about the Figures already in their caption.
Moreover, it would be very useful and it is suggested a wider discussion about the novelty of the authors method in the introduction and in the concluding section of the paper (which could be named “Discussion and conclusions”).
Comments on the Quality of English Language
The paper addresses an interesting topic in the field of automatic control systems, namely, the development of a semi-active suspension system for vibrations mitigation in electric vehicles. It is shown that the authors semi-active suspension improves ride comfort compared to PSO-BPNN-PID (PID control and Particle Swarm Optimization) and meets control performance requirements under different driving conditions. Numerical simulations are used to discuss the effectiveness of the proposed solution. Comments on the paper are hereafter.
The introduction should end with a clear and concise description of the organization of the paper, in such a way that (since the beginning) the reader have a clear picture about the content of the sections, e.g.: in Section 2 “Generative adversarial network model based on scene constraints…” is introduced; in Section 3 “Experimental results…” are reported; etc. etc.
Moreover, it is a general practice to put the cited articles right after the author’s name and not after several lines, e.g.: line 42, “Hu studied the…”, should be “Hu [4] studied the…”. Similarly: line 49 should be “Zheng [5] investigated…”. The same at line 55 “Snehasagar [6]…”, line 60 “Li [7]…”, line 66 “Krishnanunni [9]…”. Please check also the subsequent lines.
Also check minor typos within the entire document, e.g.: line 13, “consider” should be “considering”; line 30, “driven” should be “driving”; line 58, “…equation, the…” should be “…equation. The…”; etc. etc.
Some sentences are a bit convoluted (although one may get the meaning), e.g., “The vertical dynamic coupling effect model of disc-type permanent magnet motors and hub motors is proposed…”. The entire sentence appears as the subject of the verb. Please consider rewriting sentences like this and check the whole document to simplify the reading.
The caption of “Figure 4. Motor excitation.” (and other short captions) could be improved. Please consider providing further information about the Figures already in their caption.
Moreover, it would be very useful and it is suggested a wider discussion about the novelty of the authors method in the introduction and in the concluding section of the paper (which could be named “Discussion and conclusions”).
Round 2
Reviewer 1 Report
Comments and Suggestions for Authors
All my comments are addressed.
Comments on the Quality of English Language
Fine
Author Response
All my comments are addressed.
Reviewer 2 Report
Comments and Suggestions for Authors
I am satisfied with the corrections made by the authors.
Author Response
I am satisfied with the corrections made by the authors.
Reviewer 3 Report
Comments and Suggestions for Authors
Most of the previous comments have been addressed. However, consider that the "simple present" is usually employed in the introduction for the description of the content of the paper. Lines 194-201 should be corrected accordingly, e.g.: "In the second section of this paper, the vertical dynamics relationship between hub motor electric vehicles and road surfaces WAS considered" (WAS should be replaced with IS). Similarly: "...input models are established", "...control strategy is constructed...", "...control strategy is analyzed...", "Finally, in the fifth section, a summary of the entire paper is provided".
Author Response
Thank you for your feedback and guidance. We appreciate your thorough review of our paper. We have made the necessary corrections to lines 194-201, as per your recommendations. The revised sentences now reflect the use of the "simple present" tense.
Change to:"In the second section of this paper, the vertical dynamics relationship between hub motor electric vehicles and road surfaces is considered. A vehicle dynamics model and various external excitation input models are established. In the third section, an intelligent algorithm control strategy is constructed to establish a semi-active suspension controller for the vehicle. In the simulation analysis of the fourth section, the actual effectiveness of the adopted control strategy is analyzed through a comparison of the results from simulated experiments. Finally, in the fifth section, a summary of the entire paper is provided."
[Page 4 ,paragraph 4 , and line 194-200.]